# Tracking the COVID-19 pandemic in Australia using genomics

Torsten Seemann [1,2,6], Courtney R. Lane[1,6], Norelle L. Sherry [1,6], Sebastian Duchene[3], Anders Gonçalves da Silva[1], Leon Caly[4], Michelle Sait[1], Susan A. Ballard[1], Kristy Horan[1], Mark B. Schultz [1], Tuyet Hoang[1], Marion Easton[5], Sally Dougall[5], Timothy P. Stinear [2,3], Julian Druce[4], Mike Catton[4], Brett Sutton[5], Annaliese van Diemen[5], Charles Alpren[5], Deborah A. Williamson [1,3,4,7] & Benjamin P. Howden [1,2,3,7 ✉]

Genomic sequencing has significant potential to inform public health management for SARS-CoV-2. Here we report high-throughput genomics for SARS-CoV-2, sequencing 80% of cases in Victoria, Australia (population 6.24 million) between 6 January and 14 April 2020 (total 1,333 COVID-19 cases). We integrate epidemiological, genomic and phylodynamic data to identify clusters and impact of interventions. The global diversity of SARS-CoV-2 is represented, consistent with multiple importations. Seventy-six distinct genomic clusters were identified, including large clusters associated with social venues, healthcare and cruise ships. Sequencing sequential samples from 98 patients reveals minimal intra-patient SARS-CoV-2 genomic diversity. Phylodynamic modelling indicates a significant reduction in the effective viral reproductive number ($R_e$) from 1.63 to 0.48 after implementing travel restrictions and physical distancing. Our data provide a concrete framework for the use of SARS-CoV-2 genomics in public health responses, including its use to rapidly identify SARS-CoV-2 transmission chains, increasingly important as social restrictions ease globally.

[1] Microbiological Diagnostic Unit Public Health Laboratory, Department of Microbiology and Immunology, The University of Melbourne at The Peter Doherty Institute for Infection and Immunity, Melbourne, VIC, Australia. [2] Doherty Applied Microbial Genomics, Department of Microbiology and Immunology, The University of Melbourne at The Peter Doherty Institute for Infection and Immunity, Melbourne, VIC, Australia. [3] Department of Microbiology and Immunology, The University of Melbourne at The Peter Doherty Institute for Infection and Immunity, Melbourne, VIC, Australia. [4] Victorian Infectious Diseases Reference Laboratory at The Peter Doherty Institute for Infection and Immunity, Melbourne, VIC, Australia. [5] Victorian Department of Health and Human Services, Melbourne, VIC, Australia. [6] These authors contributed equally: Torsten Seemann, Courtney R. Lane, Norelle L. Sherry. [7] These authors jointly supervised this work: Deborah A. Williamson, Benjamin P. Howden. ✉email: bhowden@unimelb.edu.au

The coronavirus disease (COVID-19) pandemic caused by severe acute respiratory syndrome coronavirus 2 (SARS-CoV-2) is a global public health emergency on a scale not witnessed in living memory. First reports in December 2019 described a cluster of patients with pneumonia, linked to a market in Wuhan, China[1,2]. Subsequent testing revealed the presence of a previously unknown coronavirus, now termed SARS-CoV-2, with the associated disease termed COVID-19[2].

Initial laboratory responses included early characterization and release of the viral whole-genome sequence (strain Wuhan-Hu-1) in early January 2020[2], which enabled the rapid development of reverse transcriptase-polymerase chain reaction (RT-PCR) diagnostics[3]. To date, laboratory testing has played a critical role in defining the epidemiology of the disease, informing case and contact management, and reducing viral transmission[4]. In addition to facilitating the development of diagnostic tests, whole-genome sequencing (WGS) can be used to detect phylogenetic clusters of SARS-CoV-2[5], with many laboratories now making genomic data publicly available[6,7].

For other viral pathogens, genomic surveillance has been used to detect and respond to putative transmission clusters[8,9] and to provide information on the possible source of individual cases[10]. To ensure maximal public health utility, genomics-informed public health responses require detailed integration of genomic and epidemiological data, which in turn requires close liaison between laboratories and public health agencies. Here, we combine extensive WGS and epidemiologic data to investigate the source of individual cases of COVID-19 in Victoria, Australia. This report describes the key findings from the first 1333 cases of COVID-19 in our setting and demonstrates the integration of genomics-based COVID-19 surveillance into public health responses.

## Results

**Demographic characteristics of cases.** Over the study period (January 6, 2020–April 14, 2020), there were 1333 laboratory-confirmed cases of COVID-19 in Victoria. Of these, 631/1333 (54.2%) were male, and the median age was 47 years (IQR 29–61) (Table 1). The majority of cases (827/1333, 62.0%) were identified in returning travelers, most commonly from north–west Europe and the Americas, and 360 (27.0%) in known COVID-19 contacts (Table 1). Cases in Victoria peaked in mid-March, then declined over the study period, consistent with population-level public health interventions (Fig. 1). In total, 134/1333 (10.1%) cases were acquired within Australia from an unknown source.

**Prospective viral sequencing and genomic epidemiology.** A total of 1242 samples from 1075 patients were sequenced during the study period, representing 80.7% of all cases (Fig. 2). There were no significant demographic differences between cases with and cases without included sequence data (Supplementary Table 1). Once the sequencing workflow was established, the median time from sample collection to sample receipt at the sequencing lab was 5 days (IQR 3–7 days), while the median time from sample receipt to sequence data availability was 7 days (IQR 2–12 days).

Of the 1242 samples, 1085 (87.3%) passed our predefined quality control (QC) parameters (Supplementary Data 1); after excluding duplicate patients from cases, 903 samples (68% of cases) were included in the final alignment. While the characteristics of cases with and without included sequence data were comparable, the sequenced samples meeting QC parameters were noted to have a significantly lower PCR cycle threshold (Ct) value than sequences excluded from the final alignment (median 27, IQR 22–31 versus median 36, IQR 32–38, for excluded

**Table 1 Demographic and risk factor data for Victorian COVID-19 cases to 14 April 2020.**

| Characteristic | Number (% of total) |
| --- | --- |
| *Sex* | |
| Male | 473 (53.6%) |
| Female | 410 (45.4%) |
| Unknown | 20 (2.2%) |
| *Median age (years) (IQR)* | |
| All | 46 (29–60) |
| Males | 45 (28–58) |
| Females | 46 (29–46) |
| Healthcare worker | 109 (11.9%) |
| Residence in the metropolitan region | 755 (86.5%) |
| *Putative source of acquisition* | |
| Overseas travel | 557 (61.7%) |
| Contact with a known case | 260 (28.8%) |
| Unknown | 81 (9.0%) |
| *Region of travel (for travel-associated cases, n = 557)** | |
| Oceania | 61 (10.9%) |
| North–West Europe | 230 (41.3%) |
| Southern and Eastern Europe | 41 (7.4%) |
| North Africa and the Middle East | 24 (4.3%) |
| South–East Asia | 35 (6.3%) |
| North–East Asia | 12 (2.2%) |
| Southern and Central Asia | 7 (1.3%) |
| Americas | 169 (30.3%) |
| Sub-Saharan Africa | 8 (1.4%) |
| *Sample site* | |
| Nasopharyngeal swab/nasal swab | 839 (92.9%) |
| Lower respiratory tract specimen | 13 (1.4%) |
| Unknown/other | 51 (5.7%) |

*Cases traveling to more than one region were counted more than once.

sequences; $P < 0.001$) (Supplementary Table 2 and Supplementary Fig. 2). As reported elsewhere[11,12], we found relatively little genetic variation across the genomes, with a maximum of 15 single-nucleotide polymorphisms (SNPs) observed relative to the Wuhan-1 reference (median seven SNPs, IQR 6–9).

Almost all second-level lineages from a recently proposed SARS-CoV-2 genomic nomenclature[13] were identified in the dataset (excluding lineage A.4), suggesting that Victorian samples were representative of the global diversity of published SARS-CoV-2 sequences, consistent with epidemiological findings (Fig. 3 and Table 1).

**Genomic clustering among Victorian COVID-19 cases.** In total, 737 samples belonged to a genomic cluster, representing 81.6% of the samples in the final dataset. Overall, 76 genomic clusters were identified, with a median of 5 cases per cluster (range 2–75, IQR 2–11 cases) and a median duration of 13 days (IQR 5–22 days) (Fig. 4), consistent with repeated introduction and limited subsequent local transmission. There was strong concordance between epidemiological and genomic clusters. For each epidemiologically-linked group, a median of 100% (IQR 93–100%) of cases were identified within a single dominant genomic cluster (Fig. 5). However, a genomic cluster was commonly broader than a single group of epidemiologically linked cases; for each genomic cluster a median of only 43% of cases (IQR 23–77%) were within a single dominant epidemiologically-linked group. This may indicate unrecognized or undocumented contact between cases within the same genomic cluster. However, 50/76 (66%) of genomic clusters contained multiple travelers without known epidemiological links to each other, suggesting there may also be insufficient granularity in the genomic

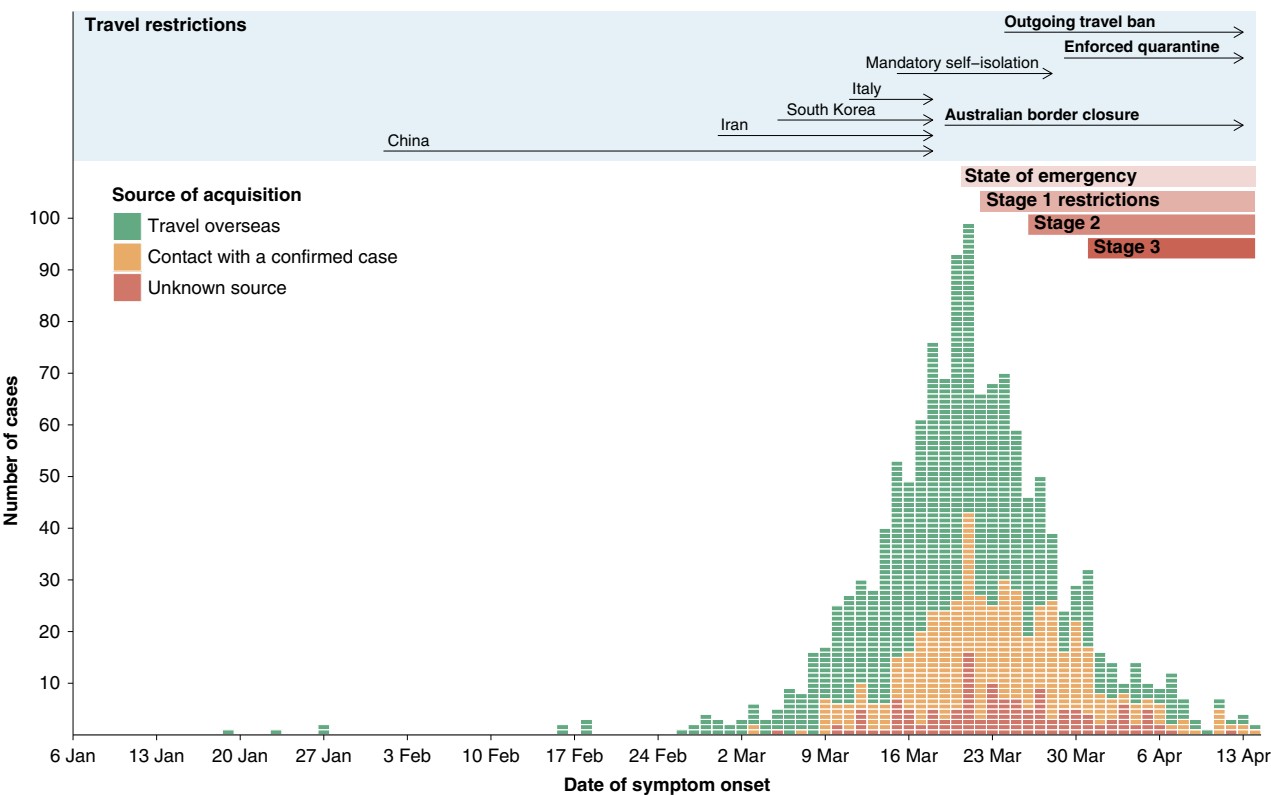

**Fig. 1 Epidemic curve of the coronavirus disease (COVID-19) cases and implementation of public health interventions.** Cases were categorized as (i) travel overseas if reporting travel in the 14 days prior to symptom onset or (ii) contact with a confirmed case if no overseas travel reported and case contact occurred within the same time period. Cases are plotted by reported date of symptom onset, or if unknown, date of initial specimen collection. The state of emergency declaration introduced a ban on large gatherings and mandatory social distancing of $4 \, m^2$ per person. Stage 1 restrictions introduced a shutdown of nonessential services, followed shortly after by early commencement of school holidays. Stage 2 restrictions expanded shutdown of nonessential services, and Stage 3 introduced an enforceable stay-at-home order and limited non-household gatherings to two people.

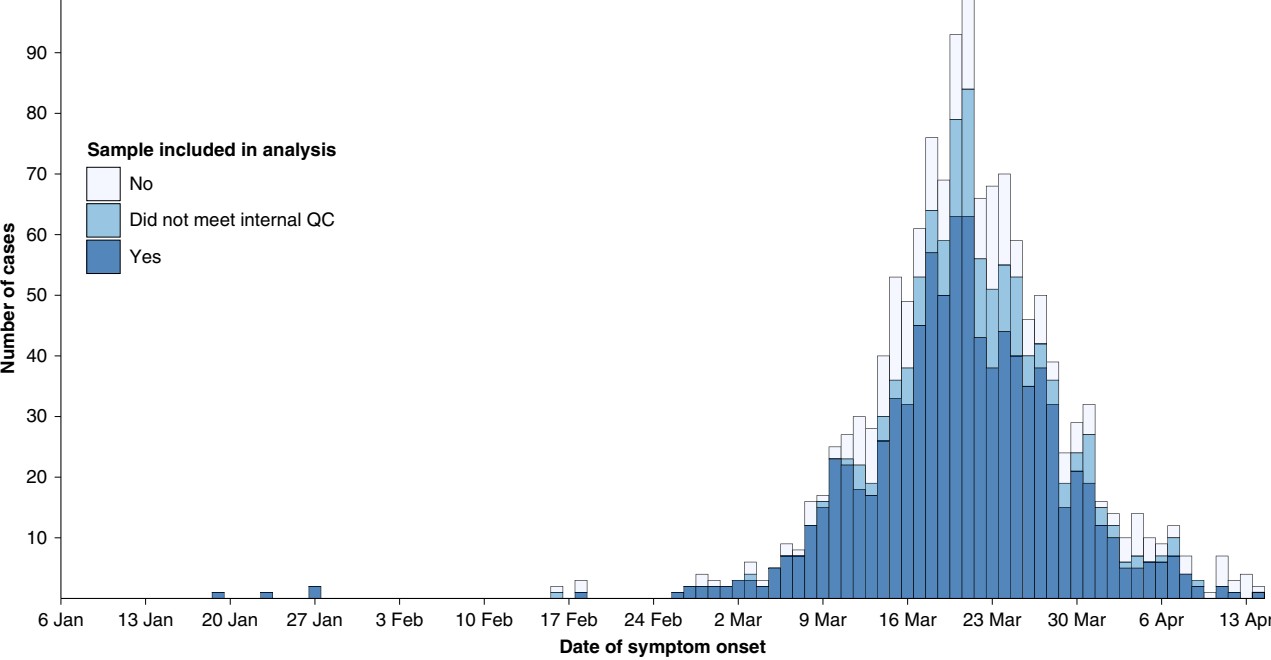

**Fig. 2 Samples included in genomic analysis.** Epidemic curve of sequenced patient samples by date of symptom onset, colored by the outcome of sequencing after quality control (QC) procedures applied. Dark blue represents successful sequencing meeting QC parameters; white represents failed sequencing; light blue represents sequences with severe acute respiratory syndrome coronavirus 2 (SARS-CoV-2) reads that did not meet internal QC parameters, but may still yield useful phylogenetic data for analysis.

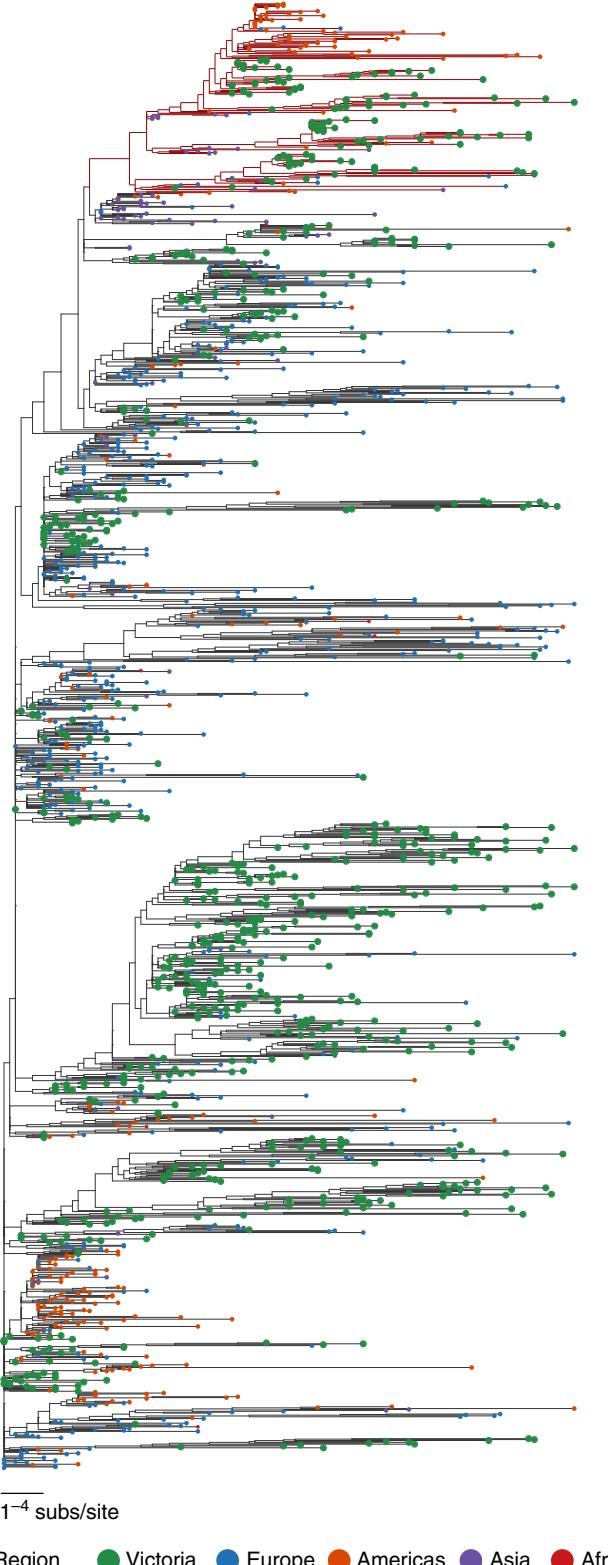

**Fig. 3 Phylogenetic tree of Victorian severe acute respiratory syndrome coronavirus 2 (SARS-CoV-2) sequences with international context.**
Maximum likelihood tree of Victorian SARS-CoV-2 sequences and a subset random selection of international sequences representing global genomic diversity, colored by region of origin. Victorian isolates, in green, have been emphasized through increased size, and represent the global diversity of the sampled SARS-CoV-2 population. Branch color represents Pangolin lineage A (red) or B (black).

clustering alone to differentiate multiple common source importations from local transmission clusters.

Of the 76 genomic clusters, 34 (45%) contained only cases reporting overseas travel; a further 34/76 clusters (45%) contained both travel-associated and locally acquired cases, with the first sampled case reporting overseas travel in 27/34 (79%) of these clusters. Using a quantitative phylogenetic approach, we inferred 193 importations (95% credible interval, CI: 180–204) that resulted in locally acquired cases, with 52 (95% CI 45–59) events leading to further transmission (i.e., transmission lineages). Transmission lineages accounted for 54% (95% CI: 50–60%) of all locally acquired cases. Although our data involve a very high sampling proportion of the total number of cases, the fact that these analyses do not include every possible infection means that these estimates represent the minimum number of transmission lineages.

Of the 81/134 sequenced cases (61%) with an epidemiologically unknown source of acquisition, 71 (88%) were identified within 24 genomic clusters, providing insight into potential sources of acquisition for these epidemiologically undefined cases. This information was provided to the genomics response team to inform public health investigation in these cases.

**Transmission clusters of public health importance**. Several genomic clusters were investigated further due to their potential to inform public health action. These included genomic clusters containing cases with no known epidemiological links, cases with multiple hypotheses for acquisition, or where putative transmissions had significant public health policy or infection control implications (Figs. 4 and 5). For example, genomic cluster 19 (total 75 cases) contained 48 cases in four epidemiological clusters associated with social venues, 7 cases from an epidemiologically unlinked health service, and 16 cases with no known epidemiological source of infection, all within a specific geographical area of metropolitan Melbourne. This genomic evidence of localized community transmission, which could not be resolved through contact tracing efforts, provided policy support for community-level social restrictions, implemented between the 22nd and 31st March (Fig. 1). No further cases were identified within this genomic cluster with onset after April 6, 2020.

Genomic analysis was also used to investigate putative interfacility transmission among four health services, which were epidemiologically linked by common healthcare workers or patients. Preliminary epidemiological analysis suggested this network comprised up to 54 indirectly linked cases; however, genomic investigations identified at least four distinct genomic clusters (clusters 9, 69, 54, and 27, Fig. 4), of which only cluster 54 contained cases associated with multiple facilities. Further investigation revealed all three cases in cluster 54 (total 15 cases) attended the same social event, along with other cases in this genomic cluster. This investigation excluded interfacility healthcare transmission and provided evidence against transmission in one health service, reducing infection control requirements and contact tracing investigations at that facility.

Among our dataset, 4/76 (5%) genomic clusters had >50% of cases associated with a cruise ship (Fig. 4). In total, 17/74 cases (23%) in these clusters had no history of overseas travel, indicating limited onwards transmission.

**Genomic assessment of intra-patient diversity**. Ninety-eight cases had more than one sample sequenced over the study period (median two sequences per case, range 2–5), with a median of 10 days between first and last sample (IQR 5–13 days) (Supplementary Table 2 and Supplementary Fig. 3). The median intra-patient pairwise SNP distance was 0 (range 0–18), compared to a

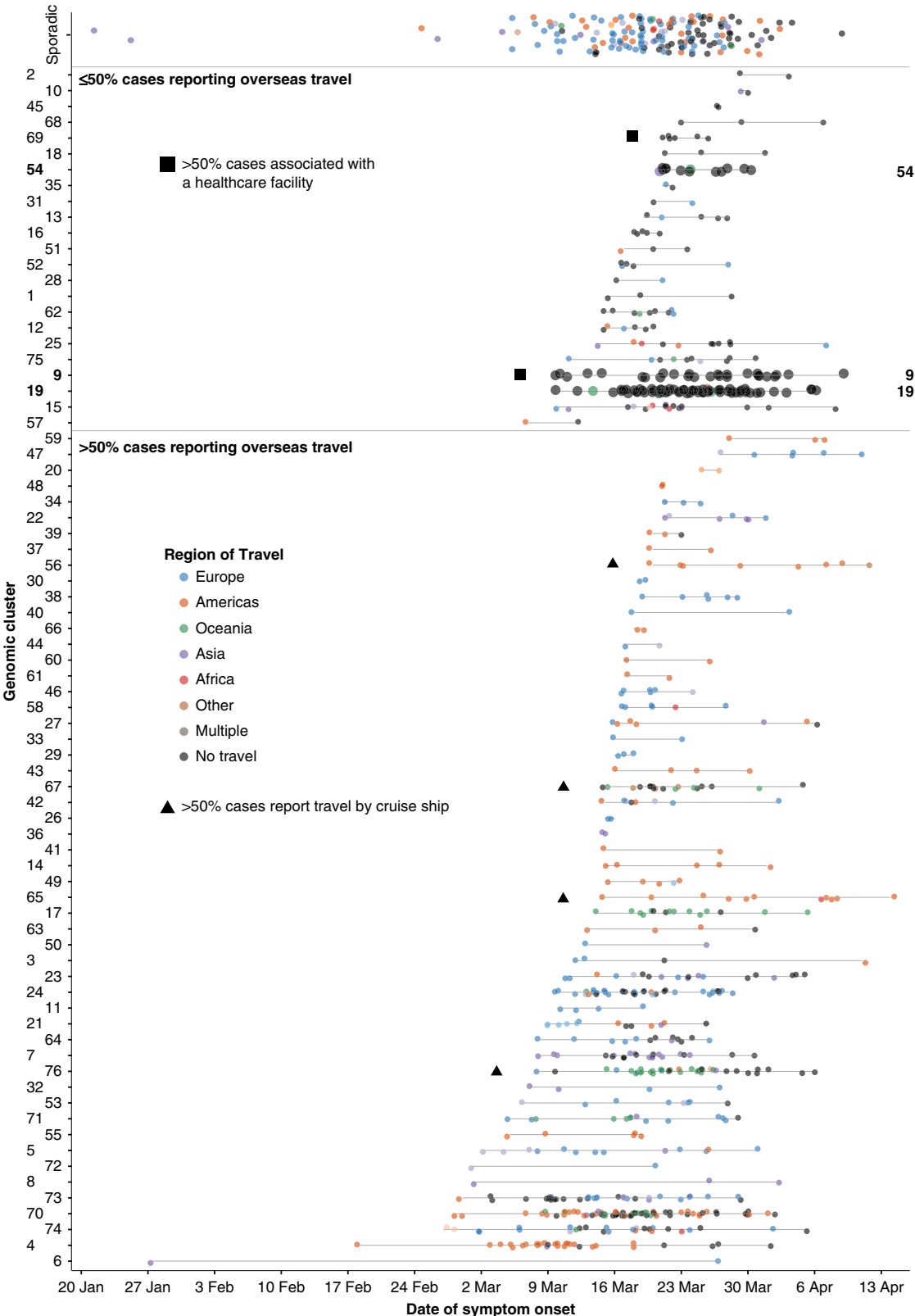

**Fig. 4 Timeline and key epidemiological features of Victorian severe acute respiratory syndrome coronavirus 2 (SARS-CoV-2) genomic clusters.** Each case in a genomic cluster is represented by a dot, colored by location of travel. Cases are plotted by onset date on the *X* axis, and genomic cluster on the *Y* axis. Genomic clusters discussed in the text are enlarged and marked with their cluster number to the right.

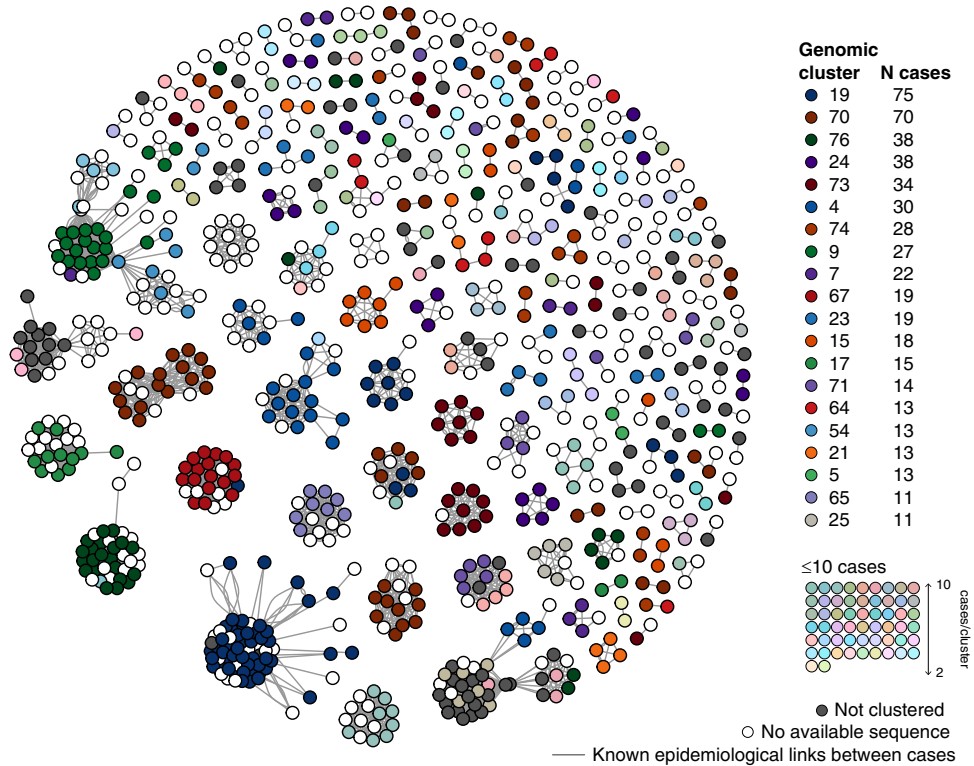

**Fig. 5 Network analysis demonstrating concordance between genomic and epidemiological clusters.** Each filled dot (node) represents a Victorian coronavirus disease (COVID-19) case with a documented epidemiological link to another Victorian case. Edges (links) between nodes (cases) represent each epidemiological link. Cases are placed closer to each other within the network as the density (number) of linkages between them increases, with cases in the same epidemiological cluster forming a spatially distinct group. Cases are colored by genomic cluster; cases without a sequence included in primary analysis are colored white.

median inter-patient pairwise SNP distance of 11 SNPs (range 0–27 SNPs). Three intra-patient pairs were outliers, having pairwise SNP distances of 7, 9, and 18 SNPs, placing the samples from each pair in a different cluster. The time between collection dates for outlier pairs (2, 12, and 6 days, respectively) was similar to non-outliers (median 8 days, IQR 3–13 days). On manual inspection, at least one sequence from each pair was found to have more ambiguous or missing base calls than the rest of the dataset, potentially contributing to the high number of intra-patient SNPs.

In order to further assess the reproducibility of SARS-CoV-2 sequencing, duplicate sequencing of the same clinical sample was also performed across different sequencing runs for ten samples, with zero SNP differences detected between consensus sequences.

**Estimating the effect of public health interventions.** Bayesian phylodynamic analyses estimated the time to the most recent ancestor of the 903 samples in December 2019 (95% credible interval, CI: 18th December to 30th December), with an evolutionary rate of $1.1 \times 10^{-3}$ substitutions/site/year, consistent with the observed diversity of phylogenetic lineages in these data.

The birth–death skyline model suggested a considerable change in $R_e$ around 27th March (CI: 23rd–31st March). Prior to 27th March, the estimated $R_e$ was 1.63 (CI: 1.45–1.8), with a subsequent decrease to 0.48 (CI: 0.27–0.69) after this time (Fig. 6). Our estimated $R_e$ prior to the 27th March implied an epidemic doubling time of 11 days (CI: 8.3–14.4 days). The posterior distribution for $R_e$ does not include one after this time, supporting a decrease in SARS-CoV-2 incidence after this time, generally consistent with results of epidemiologic modeling[14].

From the model, the sampling proportion parameter (the probability of successfully sequencing an infected case) after the identification of the first case in Victoria was estimated at 0.88 (CI: 0.7–1.0). This estimate is consistent with intensive sequencing efforts in Victoria, and is in accordance with the proportion of samples obtained for sequencing from cases in Victoria (1075/1333; 80.7%).

## Discussion

We provide a detailed picture of the emergence and limited onward spread of SARS-CoV-2 in Australia, and demonstrate how genomic data can be used to inform public health action directly. The sheer scale and rapidity of the COVID-19 pandemic have necessitated swift and unprecedented public health responses, and the high proportion of cases with associated sequence data in our study provides unique genomic insights into the effects of public health interventions on the spread of SARS-CoV-2. The genomic clusters in our dataset reflect some of the key public health and epidemiological themes that have emerged globally for COVID-19[15].

First, our genomic and modeling data demonstrate the critical role of multiple SARS-CoV-2 importations by returned international travelers in driving transmission in Australia, with travel-related cases responsible for establishing ongoing transmission lineages (each with 3–9 cases) accounting for over half of locally acquired cases. The changing origin of travel-associated clusters in our dataset (Asia, Europe, North America) is in keeping with the temporal emergence of these areas as global "hot-spots" for COVID-19, and in keeping with sequential international travel restrictions declared by the Australian Government[16]. Of note, 22% of travel-associated cases were "sporadic" (i.e., not in a

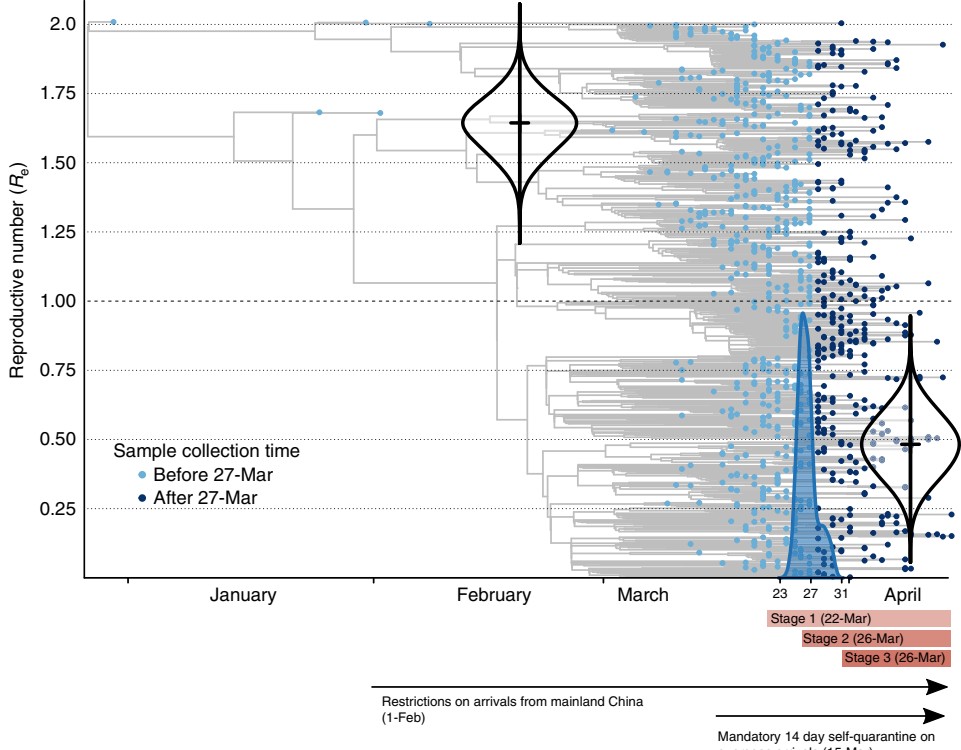

**Fig. 6 Phylodynamic estimates of the reproductive number ($R_e$).** A birth–death skyline model was fit, where $R_e$ is allowed to change at a single time point determined by the data. The X axis represents time, from the molecular estimate of the origin of the sampled diversity, around late December 2019 (95% credible interval, CI: 18th December to 30th December) to the date of the most recently collected genome in 13 April. The blue shows the posterior distribution of the timing of the most significant change in $R_e$, around 27 March (CI: 23–31 March). The Y axis represents $R_e$, and the violin plots show the posterior distribution of this parameter before and after the change in $R_e$, with a mean of 1.63 (CI: 1.45–1.80) and 0.48 (CI:0.27–0.69), respectively. The phylogenetic tree in the background is a maximum clade credibility tree with the tips colored according to whether they were sampled before or after March 27th.

genomic cluster), providing genomic evidence for the positive effects of widespread public health messaging and self-isolation requirements for returning travelers. Moreover, we identified four genomic clusters that were associated with cruise ship passengers either returning to or disembarking in Melbourne. Throughout the global COVID-19 pandemic, cruise ships have been identified as "amplification vessels" for COVID-19, with onward seeding into ports[17]. In Victoria, returning cruise ship passengers were quarantined on arrival, and our genomic data suggest only minimal onward transmission of infection from cruise ship passengers, supported by limited numbers of non-travel-associated cases in these clusters, highlighting the effectiveness of local containment measures for this high-risk population group.

Second, consistent with the impact of COVID-19 in healthcare facilities in other settings, we identified a large genomic cluster of SARS-CoV-2 in a healthcare facility in Melbourne, with cases identified in patients and staff. Although genomics has been used extensively for infection control purposes in other pathogens, our data highlight the utility of genomics for SARS-CoV-2 infection control, with potential applications in monitoring the effectiveness of local policies for identifying high-risk patients, and in assessing the effectiveness of personal protective equipment (PPE). Applying genomics in healthcare settings is particularly important in the context of high reported rates of nosocomial acquisition of COVID-19 in other settings, with associated fatalities[18,19].

Third, prior to the implementation of social restrictions in Victoria, we identified a large genomic cluster (the largest in our dataset, comprising 75 cases) associated with several social venues

in metropolitan Melbourne. Whilst the absence of multiple large clusters precluded quantitative modeling of the effect of social restrictions on local transmission, the observation of such a large genomic cluster, associated with leisure activities in this case, provides some justification for the unprecedented population-level social restrictions in our setting. Further genomic support for the effectiveness of social restrictions is provided by our phylodynamic analysis, which demonstrates a decrease in $R_e$, after the introduction of stage 3 restrictions (including mandatory quarantine in hotels for overseas returnees), from 1.63 to 0.48. The reduction in $R_e$ supports a decrease in disease incidence after the introduction of social restrictions, broadly in keeping with recent epidemiological modeling, suggesting a decrease in $R_e$ in Victoria around mid-March[14]. The differences between epidemiological and genomic modeling may be due to differences in underlying models and the expected lag between demographic processes and their effect on the molecular variation, even for rapidly evolving pathogens[20].

A major strength of our study is that we were able to sequence samples from approximately 80% of all cases in Victoria, facilitated by the centralized nature of public health laboratory services in our setting. The high proportion of sequenced cases allowed us to address very specific queries from a public health perspective (e.g., whether case X belongs to cluster Y), enabling enhanced contact tracing when there was uncertainty around epidemiological information. The key to this effort was high-throughput sequencing using an amplicon-based approach, which allowed us to process a large number of samples in a short period of time. Stringent QC to

ensure only high-quality consensus sequences entered the final alignment was particularly important when considering the minimal diversity in SARS-CoV-2 sequence data used to infer genomic clusters[11,12]. While the use of a predefined Ct value to select samples for SARS-CoV-2 genomic sequencing could be considered[21], our use of QC parameters, rather than a Ct value, enabled the inclusion of additional samples for genomic analysis, with some samples with Ct values up to 40 being successfully included.

Of further note was our assessment of intra-patient diversity, representing the largest analysis of intra-patient SARS-CoV-2 diversity to date. Whilst our analysis of intra-patient variation from consensus sequences (rather than raw reads) is a preliminary approach at this stage, our observation of minimal intra-patient SARS-CoV-2 diversity is in keeping with other recent findings[22], and provides additional evidence for the reproducibility of our sequencing and analysis. Further exploration of intra-host diversity is merited, but first requires a more in-depth understanding of the signatures of RNA degradation and other processes that could be introducing variation into SARS-CoV-2 sequence data in order to avoid known biases in intra-host diversity analyses[23].

In summary, we provide detailed genomic insights into the emergence and spread of SARS-CoV-2 in Australia and highlight the effect of public health interventions on the transmission of SARS-CoV-2. Through a combination of rapid public health responses, extensive diagnostic testing, and collective social responsibility, Australia has successfully navigated the first wave of the COVID-19 pandemic. As social restrictions inevitably ease, the role of genomics will become increasingly important to rapidly identify and "stamp out" possible transmission chains. Our data provide a framework for the future application of genomics in response to COVID-19.

## Methods

**The setting, data sources, and COVID-19 genomics response group.** In the State of Victoria, Australia (population ~6.24 million), all samples positive for SARS-CoV-2 by RT-PCR are forwarded to the Doherty Institute Public Health Laboratories[24] for confirmation and genomic analysis. We conducted a retrospective, observational study of all patients in Victoria with confirmed COVID-19 with a diagnosis prior to April 14, 2020, including collection of detailed demographic and risk factor information on each case. Epidemiological clusters were defined by investigating officers at DHHS as those that included two or more cases identified with a common exposure, such as a workplace, healthcare facility, or social venue, excluding households. Epidemiologically linked cases were defined as those within the same epidemiological cluster or where contact was otherwise identified through contact tracing.

To rapidly implement SARS-CoV-2 genomic analysis into local public health responses, a COVID-19 genomics response team was convened, including representatives from the state health department, virology laboratory, and public health genomics laboratory (genomic epidemiologist, bioinformaticians, and medical microbiologists). Laboratory and bioinformatic workflows were developed to support large-scale rapid processing of samples, enabling genomic sequencing and bioinformatic analysis of 96 samples in an approximately 45-h time period. The response team held online meetings to enable interactive reporting of genomic epidemiological analyses and facilitate the rapid translation of genomic findings into public health responses.

**Genomic sequencing and bioinformatic analysis.** Detailed methods are provided in the Supplementary Methods. In brief, extracted RNA from SARS-CoV-2 RT-PCR-positives samples underwent tiled amplicon PCR using both ARTIC version 1 and version 3 primers (Supplementary Data 2)[25] using published protocols[26], and Illumina sequencing. Reads were aligned to the reference genome (Wuhan-Hu-1; GenBank MN908947.3) and consensus sequences generated. We applied quality control checks on consensus sequences, requiring ≥80% genome recovered, ≤25 SNPs from the reference genome, and ≤300 ambiguous or missing bases for sequences to "pass" QC (Supplementary Fig. 1).

For phylogenetic analysis, a single sequence was selected per patient, and genomic clusters were defined as two or more related sequences using Cluster Picker[27]; an initial threshold of 70% bootstrap support value was used to split the tree into sub-trees for ease of computation. Clusters were then identified as those

with at least 95% bootstrap support, a maximum pairwise distance of 0.0004. Pairwise distances were calculated using the "valid" algorithm (considering only the A, C, G, and T bases). In addition, recently proposed lineages were also determined for the dataset[13]. Intra-patient sequence variability was assessed by comparing consensus sequences from different samples from the same patient (Supplementary Methods). We estimated the relative contribution of SARS-CoV-2 importation events to locally acquired cases and ongoing transmission in Victoria, by analyzing the complete alignment in BEAST2.5[28], approximating the posterior distribution, extracting 1000 trees, and inferring a number of statistics using NELSI[29] (Supplementary Methods).

**Phylodynamic analyses to estimate population parameters.** We conducted Bayesian phylodynamic analyses using the 903 genome samples from Victoria, sampled between 25 January and 14 April 2020 using BEAST2.5 (Supplementary Methods)[28]. We calibrated the molecular clock using sample collection times to estimate the evolutionary rate and timescale. To infer epidemiological dynamics, we considered a range of models including the coalescent exponential, constant birth–death and the birth–death skyline[30]. The coalescent exponential and constant birth death assume a constant reproductive number ($R_e$), such that they do not infer potential changes in this parameter due to due to government-enforced interventions, for example. Thus, we focused on the birth–death skyline model that allows for piecewise changes in the epidemiological parameter over time. Our configuration of this model consisted of two intervals for $R_e$, with the interval time co-estimated in the analysis. Although the model allows the inclusion of more time intervals, our aim was to assess the single time point with the strongest evidence for a change in $R_e$ as a means to determining whether travel and social distancing restrictions had an effect on this parameter. As such, the interval time corresponds to the date with the strongest evidence for a change in $R_e$. We assumed a duration of infection of 9.68 days to match independent epidemiological estimates reported by local mathematical modeling data[14].

**Statistical analysis.** Associations between categorical data were made using a chi-squared test, and differences in non-normally distributed numerical data using the Wilcoxon rank-sum test. All statistical analyses were performed using R (v.3.6.3). Details of additional statistical tests for modeling work can be found in Supplementary Methods.

**Ethics and study oversight.** Data were collected in accordance with the Victorian Public Health and Wellbeing Act 2008. Ethical approval was received from the University of Melbourne Human Research Ethics Committee (study number 1954615.3). All authors vouch for the integrity and completeness of data and analyses.

**Reporting summary.** Further information on research design is available in the Nature Research Reporting Summary linked to this article.

## Data availability

Consensus sequences and Illumina sequencing reads were deposited into GenBank under BioProject PRJNA613958 (Supplementary Data 1). Additional sequence data and metadata are available at https://github.com/MDU-PHL/COVID19-paper.

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

## Acknowledgements

We thank the public health, clinical, and microbiology staff across Victoria who have been involved in the testing, clinical care, and public health responses to COVID-19. The data collected through public health officers and microbiology laboratories are critical for public health genomics investigations. We also thank Nick Loman, Jonathon Jacobs, Duncan Maccannell, and Karthik Gangavarapu for bioinformatics advice, and Josh Quick, George Taiaroa, and Sara Zufan for assistance with obtaining ARTIC primers early in the pandemic. The authors gratefully acknowledge the contributions from other laboratories to GISAID (Supplementary Data 3). The Victorian Infectious Diseases Reference Laboratory (VIDRL) and the Microbiological Diagnostic Unit Public Health Laboratory (MDU PHL) at The Doherty Institute are funded by the Victorian Government. This work was supported by the National Health and Medical Research Council, Australia (NHMRC); Partnership Grant (APP1149991), Practitioner Fellowship to B.P.H. (APP1105905), Investigator Grant to D.A.W. (APP1174555), Research Fellowship to T.P.S (APP1105525).

## Author contributions

B.P.H., T.H., B.S., A.D., M.E., D.A.W., T.Se., A.G.S., N.L.S., and C.R.L. designed the study. M.S., S.A.B., L.C., and J.D. designed and performed laboratory work. C.A., S.Do., B.S., A.D., and M.E. provided epidemiologic data and implemented into public health practice. T.Se., A.G.S., K.H., and M.B.S. wrote code, designed bioinformatic pipelines, and performed bioinformatic analyses. C.R.L. collated the data, designed, and performed epidemiologic analyses. S.Du. and A.G.S. designed and performed phylodynamic modeling. T.Se., C.R.L., N.L.S., A.G.S., D.A.W., and B.P.H. wrote the paper. S.Du., S.A.B., K.H., T.P.St., M.C., and C.A. reviewed and edited the paper. C.R.L., S.Du., A.G.S., M.B.S., and N.L.S. constructed figures.

## Competing interests

The authors declare no competing interests.
