## [Peer Review File · Nature Communications]

REVIEWERS' COMMENTS:

Reviewer #1 (Remarks to the Author):

Hello,

I think the authors have done a fine job of responding to most of my initial comments.

Received: 8th Jun 20

Remarks to the Author:

*** The revised manuscript states, "We used modelling to estimate the contribution of SARS-CoV-2 importation events to locally-acquired cases and ongoing SARS-CoV-2 transmission in Victoria (Supplementary Appendix)." I would prefer the authors to elaborate on the word "modelling"... what kind of modelling? It isn't really "phylogenetic modelling" as the authors state. They do use BEAST to get a posterior sample of trees, from which I assume they pull the local vs imported infections counts, but this is not classic phylodynamics.

In the Supplement the authors describe this piece very briefly:

"We detected local transmission lineages as monophyletic groups of at least two samples from individuals with no known travel history and presumed to have been locally infected¹⁶." First, I think this approach does not appear to use a quantitative node state reconstruction (e.g. parsimony), but can the authors clarify? Also, can the authors clarify that the singletons are not counted as putative local transmissions? ***

*** The authors approach to identifying intra-host diversity is different than other approaches (which look at the raw reads) and I think just compare the consensus sequences between the two time points. The authors should note this distinction, as (if I am correct in inferring this approach from their methods) this is important. I.e. they don't report minor allele frequencies in the two different samples (from single individuals). ***

Some other questions:

1. Why were epi clusters defined as 3 or more individuals but genetic clusters defined as 2 or more individuals?

Details that I would prefer to see in the main article include:

2. How the clusters identified in ClusterPicker, i.e. what the thresholds for inclusion are. These are listed in the Supplement as "ClusterPicker (v1.2.3, options "70.0 95.0 0.0004 15")" but these need to be more completely described in terms of node support and branch lengths.

*** The authors have addressed this concern and added cluster identification details to the main manuscript. I'm not sure why they included a "maximum cluster size" threshold, though. ***

3. Why the authors chose the birth-death skyline model, and why 2 intervals for R_e . These details are included in the Supplement but I always prefer the reasoning for these choices to be explained

fully in the main document.

*** The authors have addressed this concern. ***

4. I would like more information related to this statement, "Included sequences had a significantly lower PCR cycle threshold (Ct) value than sequences excluded from the final alignment (median 27, IQR 22-31 vs. median 36, IQR 32-38, for excluded sequences; $P < 0.001$,)"

Can the authors please clarify in the main article that they mean excluded via the quality control steps? Can they discuss whether they expect these exclusions to impact their findings?

*** I didn't see any comments addressing this concern. ***

5. The authors state and infer the following about transmission clusters and social distancing, "We identified a large genomic cluster (the largest in our dataset, comprising 75 cases) associated with several social venues in metropolitan Melbourne. This finding demonstrates the propensity for chains of SARS-CoV-2 transmission throughout urban areas associated with leisure activities and provides additional justification for the unprecedented population-level social restrictions in our setting."

This seems to be an entirely anecdotal result and inference only, and should not be considered a model-based or quantitative result that demonstrates the impact of social restrictions.

*** The authors have revised this section and I think it is much improved. ***

RESPONSES TO REVIEWER'S COMMENTS
"Tracking the COVID-19 pandemic in Australia using genomics" by Seemann et al.
Revision #2

REVIEWERS' COMMENTS:

Reviewer #1 (Remarks to the Author):

Hello,

I think the authors have done a fine job of responding to most of my initial comments.

Received: 8th Jun 20

Remarks to the Author:

*** The revised manuscript states, "We used modelling to estimate the contribution of SARS-CoV-2 importation events to locally-acquired cases and ongoing SARS-CoV-2 transmission in Victoria (Supplementary Appendix)." I would prefer the authors to elaborate on the word "modelling"... what kind of modelling? It isn't really "phylodynamic modelling" as the authors state. They do use BEAST to get a posterior sample of trees, from which I assume they pull the local vs imported infections counts, but this is not classic phylodynamics.

In the Supplement the authors describe this piece very briefly:

"We detected local transmission lineages as monophyletic groups of at least two samples from individuals with no known travel history and presumed to have been locally infected¹⁶." First, I think this approach does not appear to use a quantitative node state reconstruction (e.g. parsimony), but can the authors clarify? Also, can the authors clarify that the singletons are not counted as putative local transmissions? ***

RESPONSE: The reviewer brings up an important point. We clarified this in the main text and the Supplementary Information. To avoid confusion, we refer to our approach as "Quantification of importation events", rather than phylodynamic modelling. Briefly, our approach follows the definition of transmission lineages by the COG-UK Consortium (<https://virological.org/t/preliminary-analysis-of-sars-cov-2-importation-establishment-of-uk-transmission-lineages/507>) and used elsewhere (Geoghegan et al. 2020 MedRxiv). In the present version of the manuscript we explain that we considered a set of trees from the posterior distribution to account for phylogenetic uncertainty. Although it is a quantitative approach, it does not consist of ancestral state reconstruction or explicit modelling of importation events. Singletons are not counted as putative local transmission events, such that our estimates are conservative (Page 5, line 73).

Intra-person diversity:

Intra-patient diversity seems to be assessed by temporal variation in the consensus sequences, rather than using the raw reads to identify variations at the same time points. The authors have access to the raw reads, correct? I think these should be analyzed to get inferences about intra-person diversity, in addition to longitudinal samples.

*** The authors approach to identifying intra-host diversity is different than other approaches (which look at the raw reads) and I think just compare the consensus sequences between the two time points. The authors should note this distinction, as (if I am correct in inferring this approach

from their methods) this is important. I.e. they don't report minor allele frequencies in the two different samples (from single individuals). ***

RESPONSE: We have further clarified this distinction in both the manuscript and Supplementary Methods, noting that *consensus* sequences were compared, rather than raw reads (Page 13, line 260). We have also added a further explanatory note in the Supplementary Methods, noting that our concerns about RNA degradation and other sequencing artefacts may potentially introduce minor allelic variants which do not reflect what is actually occurring in the patient. We believe that RNA degradation may be particularly significant in this dataset, as it includes samples from numerous laboratories which have not been stored optimally prior to arriving at the sequencing laboratory, and in many cases, time to sequencing has been prolonged. We are exploring this further, but this falls outside the scope of this work.

Some other questions:

1. Why were epi clusters defined as 3 or more individuals but genetic clusters defined as 2 or more individuals?

Details that I would prefer to see in the main article include:

2. How the clusters identified in ClusterPicker, i.e. what the thresholds for inclusion are. These are listed in the Supplement as "ClusterPicker (v1.2.3, options "70.0 95.0 0.0004 15")" but these need to be more completely described in terms of node support and branch lengths.

*** The authors have addressed this concern and added cluster identification details to the main manuscript. I'm not sure why they included a "maximum cluster size" threshold, though. ***

RESPONSE: We have reviewed the statement regarding maximum cluster size. It is unclear from the ClusterPicker documentation what effect this parameter has on the algorithm. We experimented with values from 0-100 for maximum cluster size, with no differences in the outcomes. This has now been removed from the manuscript.

3. Why the authors chose the birth-death skyline model, and why 2 intervals for Re. These details are included in the Supplement but I always prefer the reasoning for these choices to be explained fully in the main document.

*** The authors have addressed this concern. ***

4. I would like more information related to this statement, "Included sequences had a significantly lower PCR cycle threshold (Ct) value than sequences excluded from the final alignment (median 27, IQR 22-31 vs. median 36, IQR 32-38, for excluded sequences; $P < 0.001$.)"

Can the authors please clarify in the main article that they mean excluded via the quality control steps? Can they discuss whether they expect these exclusions to impact their findings?

*** I didn't see any comments addressing this concern. ***

RESPONSE: In the first revision, we added the qualifier excluded via the QC steps to clarify that we meant excluded via the QC steps:

“Included sequences (i.e. those meeting QC parameters) were noted to have a significantly lower PCR cycle threshold (Ct) value than sequences excluded from the final alignment...”.

To make it clearer that we are describing the characteristics of the samples failing sequencing QC, we have modified again:

“Whilst the characteristics of cases with and without included sequence data were comparable, the sequenced samples meeting QC parameters were noted to have a significantly lower PCR cycle threshold (Ct) value than sequences excluded from the final alignment...” (Page 4, line 44-45).

Additionally, in Supplementary Table 1, we also demonstrated that there were no significant differences in demographics or putative source of acquisition between the cases with sequence data available, and those without sequence data available.

5. The authors state and infer the following about transmission clusters and social distancing, "We identified a large genomic cluster (the largest in our dataset, comprising 75 cases) associated with several social venues in metropolitan Melbourne. This finding demonstrates the propensity for chains of SARS-CoV-2 transmission throughout urban areas associated with leisure activities and provides additional justification for the unprecedented population-level social restrictions in our setting."

This seems to be an entirely anecdotal result and inference only, and should not be considered a model-based or quantitative result that demonstrates the impact of social restrictions.

*** The authors have revised this section and I think it is much improved. ***